# A Lagrangian analysis of the present-day sources of moisture for major ice-core sites

A. Drumond[1], E. Taboada[2], R. Nieto[1], L. Gimeno[1], S.M. Vicente-Serrano[3], J.I. López-Moreno[3]

[1]EPhysLab, Facultade de Ciencias, Universidade de Vigo (UVIGO-CSIC), Ourense, 32004, Spain
[2]Facultade de Ciencias, Universidad de La Laguna, La Laguna, 38200, Spain
[3]Instituto Pirenaico de Ecología, Consejo Superior de Investigaciones Científicas (IPE-CSIC), Zaragoza, 38200, Spain

*Correspondence to*: A. Drumond (anitadru@uvigo.es)

**Abstract.** A Lagrangian approach was used to identify the moisture sources for fourteen ice-core sites located worldwide for the period 1980-2012. The sites were classified into three domains: Arctic, Central (Andes, Alps and Kilimanjaro), and Antarctic. The approach was used to compute budgets of evaporation minus precipitation by calculating changes in the specific humidity along 10-day backward trajectories. The results indicate that the oceanic regions around the subtropical high-pressure centers provide most of moisture, and their contribution varies throughout the year following the annual cycles of the centers. For the Arctic domain, the sources lie in the subtropical North Atlantic and Pacific. The subtropical south Atlantic, Indian and Pacific provide moisture for the Antarctic domain. The sources for South America are the Atlantic and southern Pacific, for Europe the sources are in the Mediterranean and the north Atlantic, and for Asia the sources are the Indian Ocean and the Arabian Sea.

## 1 Introduction

The most successful reconstruction of past climate has been due to the fact that stable water isotopes are conserved in ice cores (e.g. Jouzel et al., 1982; Dansgaard et al., 1993). The isotopic composition of precipitation, in deuterium, oxygen-18 and oxygen-17, depends on the climatic conditions prevailing in the oceanic regions where it originates (i.e. the sources), mainly the sea surface temperature and the relative humidity of air (Jouzel et al., 2013). The deuterium excess may be seen as a control parameter of air moisture trajectory history, because it will change when the trajectory moves over regions presenting different moisture conditions (e.g. sea/land, dry/wet land). Deuterium excess variations have been traditionally associated to changes in the temperature of the oceanic sources, but nowadays it is thought to be also related with changes in the relative humidity of the air in the source region (Pfahl and Sodemann, 2014). In any case, deuterium excess variations in ice cores may reflect past changes in the climate conditions of the oceanic sources (e.g. Masson-Delmotte et al., 2005; Steffensen et al., 2008). This information can be very useful to understand changes linked to modifications in the atmospheric circulation because the position and conditions of the moisture sources for precipitation could be altered (e.g. Masson-Delmotte et al., 2005). That is why the knowledge on the transport of moisture is crucial for the interpretation of

stable isotopes in precipitation and in paleo-archives through the understanding of the physical climatic processes involved (Sodemann and Zubler, 2009).

The analysis of the moisture sources for regions where ice-core are settled has also importance in terms of water resources. In the present climate, the polar ice sheets of Greenland and Antarctica are the largest freshwater reservoirs, and exert some control on global sea levels (Sodemann et al., 2008). According to Sodemann et al. (2008), changes in the mass balance of these regions can affect the salinity of the surrounding oceans. Outside the polar latitudes, ice-cores must be extracted from glaciers located at high topographic elevations. In general these regions are the headwaters of rivers, and the winter precipitation stored in these glaciers is released during the year, contributing to river runoff and water resources (Sodemann and Zubler, 2009).

The investigation of sources of moisture may be conducted using different techniques such as via the assessment of the atmospheric water balance (Peixoto and Oort, 1992), general circulation models (Delmotte et al., 2000), or trajectories (Rejimer et al., 2002). As the aim of this research is not a comparison among techniques, readers are suggested to consult Gimeno et al (2012), who summarized and compared the general techniques to identify moisture sources. During recent years, the use of Lagrangian trajectories methods has become popular for diagnosing the transport of moisture and for determining the origin of moisture that precipitates in particular regions because these methods allow a detailed budget of moisture along the trajectories (Gimeno et al., 2012). Sodemann and Stohl (2009) and Nieto et al. (2010) applied Lagrangian diagnostic schemes to identify the major sources of moisture for the principal Antarctic ice-core sites by back-tracking the air masses that ultimately reached these regions over a 5-year period (2000–2004). Other works have created longer back-trajectory data sets for specific ice core sites using different Lagrangian approaches. For example, Scarchilli et al (2011) investigated the precipitation events over the East Antarctic Ice Sheet using the HySPLIT Lagrangian model integrated with the ERA40 data from 1980 to 2001. Markle et al (2012) also used HySPLIT, but integrated with NCEP/NCAR data from 1979 to 2010, to investigate the synoptic variability in the Ross Sea region, Antarctica. In this study we apply the sophisticated Lagrangian diagnostic scheme previously used by Nieto et al. (2010) over short periods to present an annual and seasonal climatology of the moisture sources for fourteen of the major ice-core sites investigated worldwide for the present-day climate (1980-2012).

**2 Data and Method**

Our Lagrangian approach follows the method developed by Stohl and James (2004, 2005), that accounts for the loss and gain of moisture along air masses trajectories. For this work we use the FLEXPART V9.0 particle dispersion model fed with ERA-Interim Reanalysis data (Dee et al., 2011), the state-of-the-art reanalysis in terms of hydrological cycle. According to the analyses of Trenberth et al. (2011) and Lorenz and Kunstmann (2012), the performance of ERA-Interim in reproducing

the hydrological cycle and the water balance closure is better than ERA-40 and the newest reanalysis products Modern Era Retrospective-Analysis for Research and Applications and Climate Forecast System Reanalysis (MERRA) (Rienecker et al., 2011).

In the Lagrangian frame of reference the observer follows an individual fluid parcel as it moves through space and time. In the model the atmosphere is divided homogeneously into three-dimensional finite elements (hereafter 'particles'), each representing a fraction of the total atmospheric mass (Stohl and James, 2004). These particles may be advected backward or forward in time using three-dimensional wind taken from the meteorological data (e.g. reanalysis project) every time step, with superimposed stochastic turbulent and convective motions. The rates of increase ($e$) and decrease ($p$) of moisture ($e$-$p$) along the trajectory of each particle were calculated via changes in the specific moisture ($q$) with time ($e$-$p = m\ dq/dt$), with $m$ being the mass of the particle. Similar to the wind field, $q$ is also taken from the meteorological data. By summing ($e$-$p$) for all the particles residing in the atmospheric column over a given area A, we obtained the surface freshwater flux ($E$-$P$), where ($E$) denotes the evaporation rate and ($P$) denotes the precipitation rate per unit area. If we considered all the particles present in the atmospheric column, the results would be similar to the freshwater flux calculated via the Eulerian reference (Stohl and James, 2004). Nevertheless, the Lagrangian methodology allows us to identify the particles affecting a particular region and to calculate the surface freshwater flux ($E$-$P$) using information on the trajectories of these selected particles. In this way, a comparison between the ($E$-$P$) fields generated by the Eulerian and the Lagrangian (when selecting the particles affecting a particular region) approaches may indicate how similar are the moisture budget associated with the tracked particles to the freshwater flux observed at surface. A detailed description of this methodology is presented by Stohl and James (2004).

The FLEXPART data set used in this study was provided by a global experiment in which the entire global atmosphere was divided into approximately 2.0 million 'particles'. The tracks were computed using ERA-Interim reanalysis data at six-hour intervals, at a 1° horizontal resolution and at a vertical resolution of 61 levels from 0.1 to 1000 hPa. The analysis covered the period from January 1980 to December 2012, and the number of particles in the globe per time step was kept constant. As stressed by Gimeno et al. (2013), the FLEXPART model requires consistent high-quality data for wind and humidity, precluding its application to older reanalysis data (pre-1979), i.e., prior to the significant decrease in the errors in these variables following the inclusion of satellite data (Bengtsson et al., 2004).

30

The fourteen ice-core sites (hereafter target regions) investigated in the present study (Fig. 1 and Table 1) may be classified into three domains: the Arctic Domain (90°N – 60°N), which includes Greenland (GISP-2, NGRIP and NEEM), Alaska (Logan, Bona and Churchill), and the Franz Josef Land archipelago (Windy Dome); the Central domain (60°N – 60°S), which encompasses the South American (Huascarán and Sajama) continent, as well as Europe (Grenzgletscher), Asia

(Everest) and Africa (Kilimanjaro); and finally the Antarctic domain (60°S – 90°S), including the Byrd, EPICA DML (Dronning Maud Land) and Vostok sites. Amongst the whole set of records available, the choice of the ice cores was a first attempt to cover the geographical distribution of the major sites available worldwide and to exemplify how our methodology can be useful for ice core studies. Target regions were defined in the geographical locality (Table 1) of each ice-core site.

Except for EPICA DML (whose area was defined by a rectangle), the target regions consisted of octagons with an equivalent radius of 100 km centred on each ice-core site. The reader may see in the results that, in the general, the large-scale moisture transport patterns remain similar for sites located very close to each other (e.g. Logan and Bona-Churchill sites in Alaska) because they may be within the same or neighboring grid-points of the 1-degree reanalysis data used in this study.

Each particle identified over the target region was tracked backward in time for a period of 10 days, and its position and specific humidity ($q$) were recorded every 6 hours. While the 10-day period of tracking is somewhat arbitrary, it is about the average residence time of water vapour in the global atmosphere (Numaguti, 1999). In the time line adopted, the particles end in the ice core sites in day 0, and the previous days of the backward trajectories are day-1, day-2,… day-10. The approximate number of particles identified every time step over each one of the fourteen locations is shown in the Table 1.

Stohl and James (2004) state that the estimative of the moisture budget is valid when the number of particles per grid column of the input meteorological data exceeds the number of the layers. With this methodology, the evaporative sources and sink regions for a target region can be identified. All areas where the particles gained (lost) humidity ($E-P > 0$) ($E-P < 0$) along their trajectories towards the target region can be considered "sources of moisture" (sinks). Only regions where ($E-P) > 0$ were considered in our analysis, because they indicate those areas where air particles located within that vertical column, and

headed for their target regions, gained moisture. For each target region, a percentile criterion was applied to the climatological annual positive ($E-P$) field to define a threshold delimiting the spatial extent of the respective sources of moisture. The 95th percentile of the climatological annual positive ($E-P$) values obtained for each ice-core site delimits those regions where the air masses were likely to have picked up a large amount of moisture on their transit towards the target regions. In other words, the 95[th] percentile criteria would show the 5% grid points with the highest positive ($E-P$) values in

the annual mean map obtained for each ice core site. The same annual 95th percentile value was then applied to identify the spatial extent of the sources at seasonal scales.

The method is mostly limited by the use of a time derivative of the humidity (numerical errors associated with the temporal variations in the moisture present in a particle can be taken as moisture fluxes) (Stohl and James, 2004). In consequence, if

the meteorological data used to drive the method does not properly close the water budget, then the method may suffer from considerable inaccuracies (Gimeno et al., 2012). Other reasons for uncertainties in the calculated trajectories are the limited resolution and uncertainties in the input reanalysis data, particularly the vertical wind component, and in the interpolation of the data (Schlosser et al., 2008; Scarchilli et al., 2011). Numerical truncation errors seem to be of minor importance (Stohl et al., 2001). Average errors of approximately 20% of the travel distance may be considered typical (Stohl et al., 2001).

However, especially in areas with low meteorological data coverage, as Antarctica, trajectory errors can be larger. On one hand, Scarchilli et al. (2011) suggest that spatial uncertainties in 5-day trajectories over Antarctic are 15–30% of the total path distance, a high value probably due to the coarse grid of the meteorological model data used (2.5°horizontal resolution and at a vertical resolution of 24 levels). On the other hand, Schlosser et al. (2008) suggest that mean errors of 20% may be typical of 5-day back-trajectories over Antarctic. In spite of the uncertainties described, such random errors may cancel each other out given the large number of trajectories considered, as well as of particles found in a 1° horizontal resolution atmospheric column (Stohl and James, 2004).

The methodology applied here follows the pioneers works of Stohl and James (2004; 2005), considering the regions of (*E-P*) >0 along the trajectories selected as moisture sources and tracking all the air masses reaching the target region independently of the occurrence of precipitation events there. Other moisture sources diagnostic schemes are available (Gimeno et al.2012), such as the Lagrangian method proposed by Sodemann et al. (2008) to identify the origin of precipitation. In their approach, the cumulative moisture changes along the trajectory are also considered besides the net gain or loss at each grid point, what is necessary for quantifying the contribution of the air parcel for the precipitation in the target region. Anyway, since the purpose of the present work is to estimate the climatological moisture sources of all air masses reaching the target regions, independently of the occurrence of precipitation in the ice core sites, the use of this simple Lagrangean approach seems reasonable.

## 3 Results

Only the results for the annual and JJA and DJF periods are discussed here. The results for the transition seasons MAM and SON can be found in the suplementary material. Figures 2 and S1 shows the vertically integrated moisture flux (VIMF) calculated directly from ERA-Interim. Figures 3 and S2 are based on (*E-P*) of the tracked particles averaged over the 10-d trajectory period and redistributed on a regular 1° grid, and they provide a summary of the sources of moisture for the ice-core sites.

Figure 2 shows the VIMF and the respective divergence at an annual scale, as well as for the DJF and JJA seasons. The warm (cold) colours indicate regions of maximum divergence (convergence) of the moisture flux, which may be considered to be the evaporative (precipitating) areas. The major evaporative sources around the globe coincide with the subtropical high pressure centers observed in both hemispheres, which intensify during the respective winter seasons (DJF for the Northern Hemisphere and JJA for the Southern Hemisphere). Other regions are characterized by a strong seasonality of the VIMF divergence, such as for the monsoon domains in Asia, America, and Africa. Intense precipitation characterizes these areas during their respective summers, while divergent VIMF and evaporative conditions prevail during their respective winters.

Figure 3 summarizes the major sources of moisture for all the ice-core sites analysed at the annual scale, and for DJF and JJA. For the Arctic Domain, most of the moisture comes from latitudes 30ºN-40ºN, coinciding with the location of the subtropical high-pressure centers, and the contribution from these sources increases during the boreal winter (DJF). The

North Pacific Ocean is the major source for the sites in western North America, namely Logan (gray contour), and Bona-Churchill (yellow contour). The applicability of the Logan ice-core records to investigate the variability in the climate of the North Pacific Ocean was previously reported by Moore et al. (2002). The North Atlantic Ocean provides moisture for the Windy Dome (orange) and for the three sites in Greenland: GISP-2 (blue), NEEM (brown), and NGRIP (violet). We note that the results obtained for Greenland agree with the findings of Sodemann et al. (2008). During JJA (boreal Summer),

probably associated with the predominance of dry conditions in the northern extratropical continental regions, and with the weakening of the boreal subtropical high-pressure centers and respective moisture transport, the North American continent and the Atlantic Ocean appear to be the major sources of moisture for the Greenland sites, while the moisture for the Windy Dome mainly comes from Eurasia. Using water isotopologues as tracers, Kurita (2011) suggested that the humidity source of Arctic air masses switches in early winter from locally driven to moisture transport from lower latitudes, what agrees with

the general patterns verified in our results.

For the Central Domain, the sources of moisture were identified for ice-core sites in the Andes, Alps, Everest, and Kilimanjaro. In the Andes (South America), the Huascarán (Figure 3, dark blue) and Sajama (red) sites receive moisture from the Tropical Atlantic and the Eastern Pacific, and these sources show a strong seasonality. The contribution from the

Pacific may be due to the presence of the South Pacific subtropical high (SPSH) and the associated southerly moisture flux over the coast (Figure 2). The proximity of both ice-core sites to the Amazonian basin may explain the contribution from the Atlantic. According to Drumond et al. (2014) and the references therein, the tropical Atlantic is the major remote moisture source for the Amazon, made possible thanks to transport by the trade winds. Our results agree with the previous isotope analysis discussed by Thompson et al. (2000), who show that the dominant moisture source for Huascaran and Sajama is the

tropical Atlantic, and water vapour is advected from the east and northeast over the Amazon. As far as the seasonal changes in the Pacific are concerned, where the source expands along the coast in DJF, it is displaced northwards in JJA. These changes may be understood in terms of the migration of the SPSH northwards in JJA, when it probably reduces in terms of its influence over the areas of interest (Figure 2). In the Atlantic ocean, the contribution from the northern hemisphere in DJF may be related to the intensification of the Azores high and the associated moisture transport by the northerly trade winds

(Figure 2), an important mechanism of moisture transport for the South American Monsoon System (Drumond et al., 2014, 2008). During the inactive phase of the monsoon in JJA, the Azores high weakens and the South Atlantic Subtropical High (SASH) intensifies and migrates northwards, favoring moisture transport from the southern Atlantic Ocean to the Andes ice-core sites.

The Grenzgletscher site (Figure 3, green) in the southern Alps receives moisture mainly from the Mediterranean basin, but also receives it from the subtropical North Atlantic zone and the Mexican Gulf in DJF. The weakening of the VIMF divergence over the Azores high in JJA (Figure 2) may be related to the reduction of the contribution from the Atlantic at this time. During JJA, part of the European continent is characterised by dry conditions, and then the land surface becomes the major moisture source for the Alps. Our results corroborate those of Sodemann and Zubler (2009), who identified the moisture sources for precipitation in the Alps using a different Lagrangian approach. They found that the Southern Alps, where the ice-core site is located, receive a large proportion of their precipitation from the Mediterranean, with considerable month-to-month variability; there is a clear change from an oceanic mode during winter to a European continental mode during summer. Agreeing with our results, Mariani et al. (2014) analysed water stable isotope ratios and net snow accumulation in the Fiescherhorn (northern Alps) and the Grenzgletscher ice-core sites and also found that the two glaciers are influenced by different precipitation regimes, with the Grenzgletscher reflecting the characteristic precipitation regime south of the Alps and the Fiescherhorn, the north.

The Arabian Sea and the Indian Ocean are the major moisture sources for both Kilimanjaro in eastern Africa (Figure 3, yellow) and Everest in southern Asia (Figure 3, gray), in agreement with the major moisture sources as previously identified for these regions (Gimeno et al., 2010, 2013). It is known that the characterization of the eastern African rainfall is primarily associated with the annual migration of the Intertropical Convergence Zone (ITCZ) (Camberlin and Philippon, 2002). During the inactive phase of the Indian monsoon and the migration of the ITCZ towards the southern hemisphere in DJF, high pressure and VIMF divergence conditions are observed around India (Figure 2), and the Arabian Sea acts as the major moisture source for Kilimanjaro and Everest. However, the migration of the ITCZ into the northern hemisphere, the intensification of the Indian subtropical high and the active phase of the Indian monsoon in JJA favour the transport of moisture from the Southern Indian Ocean northwards (Figure 2). In consequence, the Indian Ocean becomes the main source for Kilimanjaro, and also provides moisture for Everest. The importance of the Indian Ocean in providing moisture to snowfall events in the Kilimanjaro was previously pointed out by Chan et al. (2008) through a moisture flux analysis. Making use of proxy indicators of relative moisture balance for the past millennium paired with long control simulations from coupled climate models, Tierney et al.(2013) also suggest that the Indian Ocean is the primary influence on East African rainfall over multidecadal and longer timescales. The isotope analysis of Aizen et al. (1996) reported the importance of the Indian Ocean, besides of the Pacific and Atlantic Oceans, in providing moisture for the northern slope of the Himalaya and southeastern Tibet. Yao et al. (2013) also reported a strong variability intra-annual in the moisture origin of the precipitation over southern Tibetan Plateau because of the influence of the Indian monsoon system over the region.

Most of the moisture for the ice-core sites in the Antarctic domain comes from the region around the austral subtropical high pressure centers (Figure 3), in good agreement with the previous findings of Nieto et al. (2010) at annual scale for a 5-year period 2000-2004. Southern Africa and the southern Atlantic are the major sources for EPICA DML (Figure 3, clear blue).

Vostok (Figure 3, pink) receives moisture from the Indian Ocean, southern Africa and southern Oceania. Byrd (Figure 3, purple) also receives moisture from southern Oceania, as well as from the Pacific Ocean. In JJA the contribution from these sources is enhanced in association with the intensification of the austral high pressure centers and the associated moisture transport. The moisture source regions appear a bit narrow in latitude compared to the results of Sodemann and Stohl (2009). These discrepancies may occur due to the differences in the methodologies. While the method applied here diagnoses the joint quantity $E$-$P$ of evaporation ($E$) and precipitation ($P$) per grid point for all air parcels reaching the target region (associate or not with precipitation), Sodemann and Stohl (2009) take the temporal sequence of moisture increases and decreases of an air parcel into account for quantitatively estimating the moisture source's contribution to the air parcel's total moisture content, and hence to the precipitation generated from it over the target.

## 4 Summary

The Lagrangian approach proposed by Stohl and James (2004) was applied together with the Era-Interim data set in order to identify the major moisture sources for the present climate (1980-2012) for fourteen of the more commonly investigated ice-core sites worldwide. The sites were classified into three domains (Arctic, Central, and Antarctic), and the fields of the vertically integrated moisture flux were also obtained in order to complement the analysis with an Eulerian perspective on the moisture transport.

The results shown in Figure 3 indicate that the oceanic regions of the subtropical high-pressure centers act as the major sources of moisture, and their contribution varies throughout the year following the annual cycles of the centers. For the Arctic domain, the main moisture sources are the subtropical North Atlantic and North Pacific. The subtropical south Atlantic, Indian and south Pacific provide moisture for the Antarctic domain. For the Central domain, the major sources of moisture for South America are the Atlantic and southern Pacific, the north Atlantic and Mediterranean for Europe, and the Indian Ocean and the Arabian Sea for Asia and Africa.

Because the main focus was to illustrate the applicability of the this Lagrangian methodology in studies of moisture transport towards ice core sites, in this work the moisture source information was not exploited with respect to ice core water stable isotope data. For example, it would be highly valuable to calculate, based on the moisture origin, the evaporation conditions (e.g. relative humidity at SST) and therefore the expected deuterium excess signal, if this would be the only driver. Such calculation would be compared with the spatial distribution of deuterium excess from different ice cores. The Lagrangian approach is not only interesting for ice core data but also for the interpretation of water stable isotope records recently obtained from continuous, in situ monitoring of surface water vapour (e.g. Steen-Larsen et al, 2013, 2015). This combination allows to both test the validity of the backtrajectory calculation using the isotopic signals, and to provide a framework to accurately investigate the meteorological drivers of isotopic fractionation.

These findings contribute to a better understanding of the moisture transport towards some of the major ice-core sites during the present climate conditions. This information is of key importance not only in paleoclimate research, given that changes in moisture origin and transport pathways can change the isotope-temperature transfer function at the basis of the interpretation of ice cores (e.g., Jouzel et al., 2003), but also for water resources management because the ice that melts from glaciers containing the ice cores may make an important contribution to runoff and water resources.

The aim of the present work focused on summarizing the climatic location of the moisture sources for the different sites selected. Nevertheless, the interannual variability of these sources and its relationship with the climatic variability modes are also information particularly interesting for the interpretation of ice core records. In addition, the use of FLEXPART with past-climate simulations data might reveal possible changes in the moisture sources associated with different atmospheric circulation patterns and moisture transport for past eras, and this information would therefore be helpful in the reconstruction of historical climates. These aspects will be analysed in future research.

**Acknowledgements**

This work was supported by the EPhysLab (UVIGO-CSIC Associated Unit). We acknowledge the support of the Spanish Government and FEDER through the "Transporte de Humedad en la Atmosfera TRAMO" (CGL-2012-35485) and the "Red de Variabilidad y Cambio Climático RECLIM" (CGL2014-517221-REDT) projects.

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

25

30

**Table 1:** Characteristics of the ice core sites analysed.

| Domain | Num. | Site | Lat. | Long. | High (m) | Max. depth (m) | Temporal coverage (year) | Number ~ of particles identified per time step | Reference |
|---|---|---|---|---|---|---|---|---|---|
| Arctic | 1 | GISP2 | 72,60°N | 38,50°W | 3200 | ~2790 | ~110000 | 220 | Meese et al. (1997) |
| | 2 | NEEM | 77,45°N | 51,07°W | 2479 | ~2540 | ~108000 | 185 | Rasmussen et al. (2013) |
| | 3 | NGRIP | 75,10°N | 42,30°W | 2917 | ~3084 | ~123000 | 185 | Andersen et al. (2004) |
| | 4 | MONTE LOGAN | 60,58°N | 140,58°W | 5340 | ~186 | ~8000 | 230 | Fisher et al. (2004) |
| | 5 | BONA-CHURCHILL | 61,40°N | 141,70°W | 4420 | ~460 | ~2500 | 215 | Mashiotta et al. (2004) |
| | 6 | WINDY DOME | 80,78°N | 63,53°E | 580 | ~315 | ~10000 | 780 | Henderson (2002) |
| Central | 7 | HUASCARÁN | 9,18°S | 78,02°W | 6048 | ~166 | ~20000 | 75 | Thompson et al. (1995) |
| | 8 | SAJAMA | 18,10°S | 68,97°W | 6542 | ~133 | ~20000 | 66 | Thompson et al. (1998) |
| | 9 | GRENZGLETSCHER | 45,92°N | 7,87°E | 4200 | ~125 | ~77-20 | 150 | Eichler et al. (2000) |
| | 10 | EVEREST | 28,02°N | 86,97°E | 6518 | ~117 | ~4000 | 70 | Hou et al. (2013) |
| | 11 | KILIMANJARO | 3,13°S | 37,58°E | 5893 | ~51 | ~11700 | 80 | Thompson et al. (2002) |
| Antarctic | 12 | BYRD | 80°S | 119°W | 1530 | ~2164 | ~100000 | 450 | Thompson et al. (1975) |
| | 13 | EPICA DML | 75°S | 0°E | 2892 | ~2774 | ~150000 | 1215 | Ruth et al. (2007) |
| | 14 | VOSTOK | 78°S | 106°E | 3488 | ~3623 | ~440000 | 790 | Petit et al. (1999) |

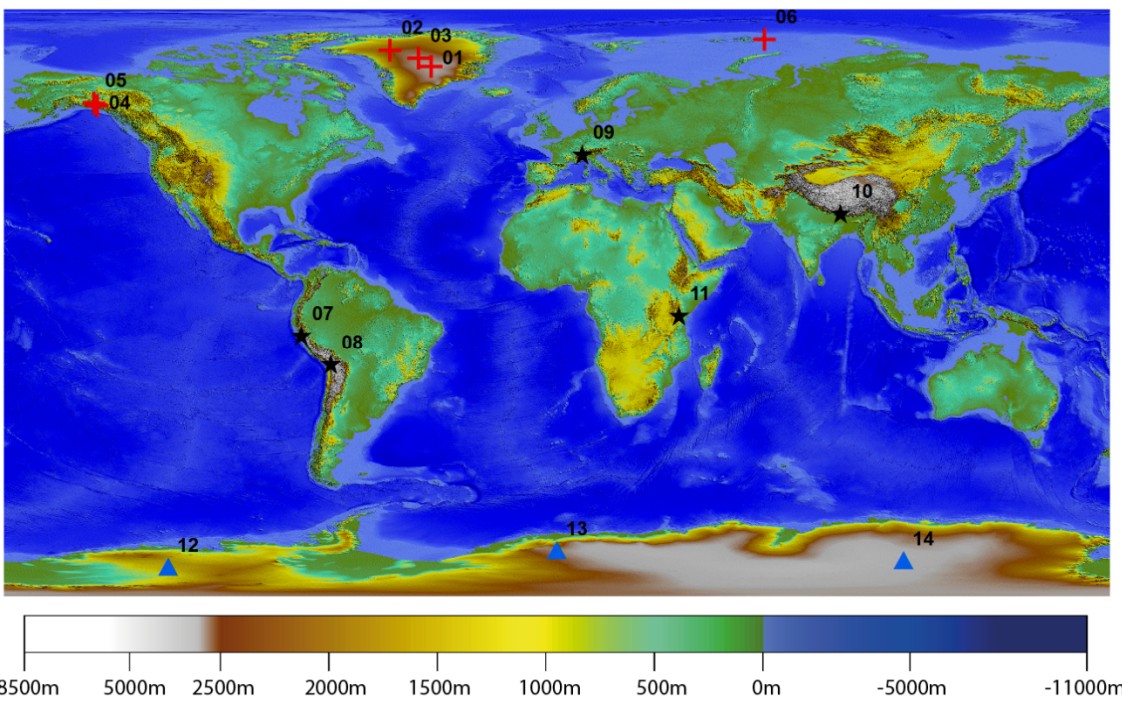

**Figure 1**: Topographic height (color bar, m) and the location of the ice-core sites. The red crosses refer to sites in the Arctic Domain, the black stars in the Central, and the blue triangles in the Antarctic Domain. The numbers are defined in Table 1.

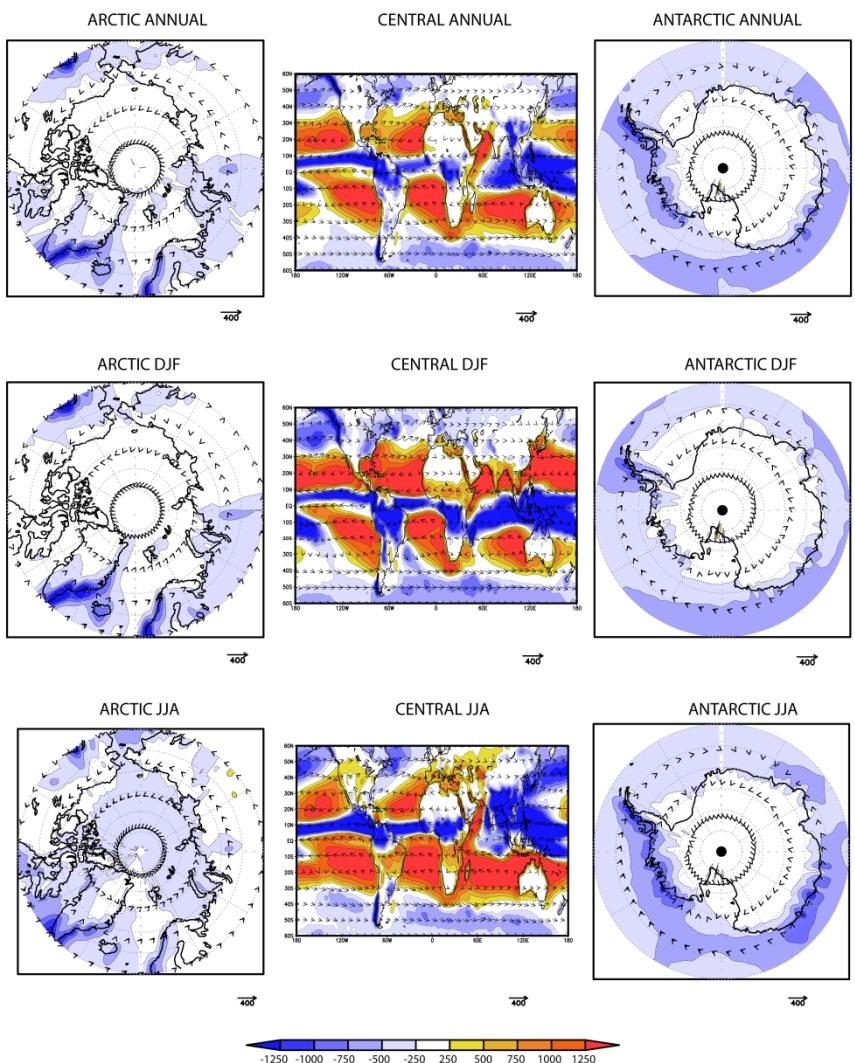

**Figure 2:** 1980-2012 vertically integrated moisture flux (vector, kg/m/s) and its respective divergence (shaded, mm/yr) in the Arctic (left-hand column), the Central (central column), and the Antarctic (right-hand column) domains for the annual (top row), DJF (central row) and JJA (bottom row) temporal means. Data: Era-Interim

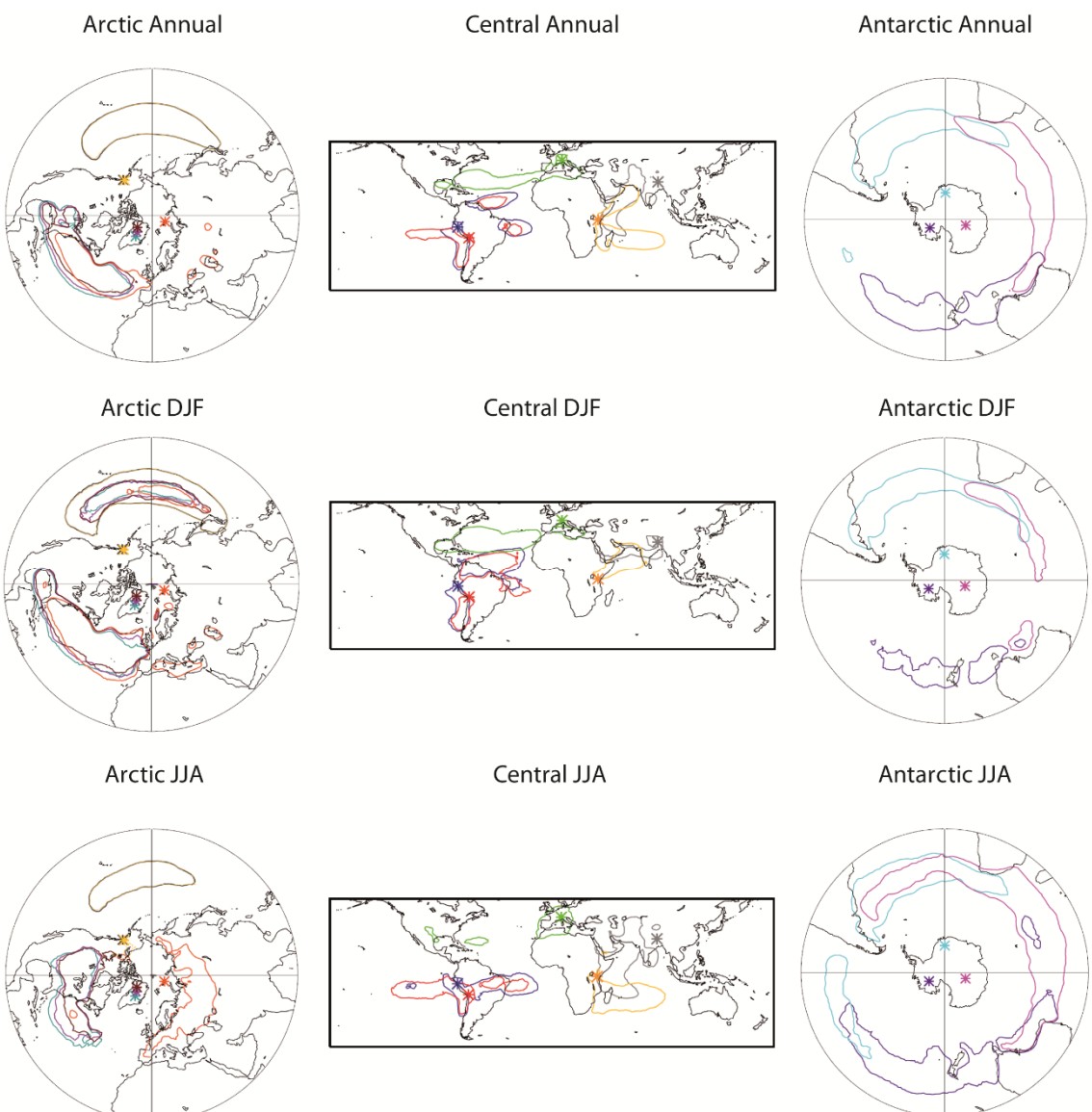

**Figure 3:** Moisture sources for the ice–core sites located in the Arctic (left-hand column), Central (central column), and Antarctic (right-hand column) domains for the annual (top line), DJF (central line) and JJA (bottom line) temporal means. The contour lines represent the 95th percentile of the positive *E-P* values in the annual mean for each ice-core Lagrangian analysis. The asterisks indicate the location of the ice-core sites investigated, represented by different colours. The colours are for the Arctic domain: Logan (gray), Bona-Churchill (yellow), Windy Dome (orange), GISP-2 (blue), NEEM (brown), NGRIP (violet); for the Central domain: Huascarán (dark blue), Sajama (red), Grenzgletscher (green), Kilimanjaro (yellow), Everest (gray); for the Antarctic Domain: Byrd (purple), DML (clear blue), and Vostok (magenta).

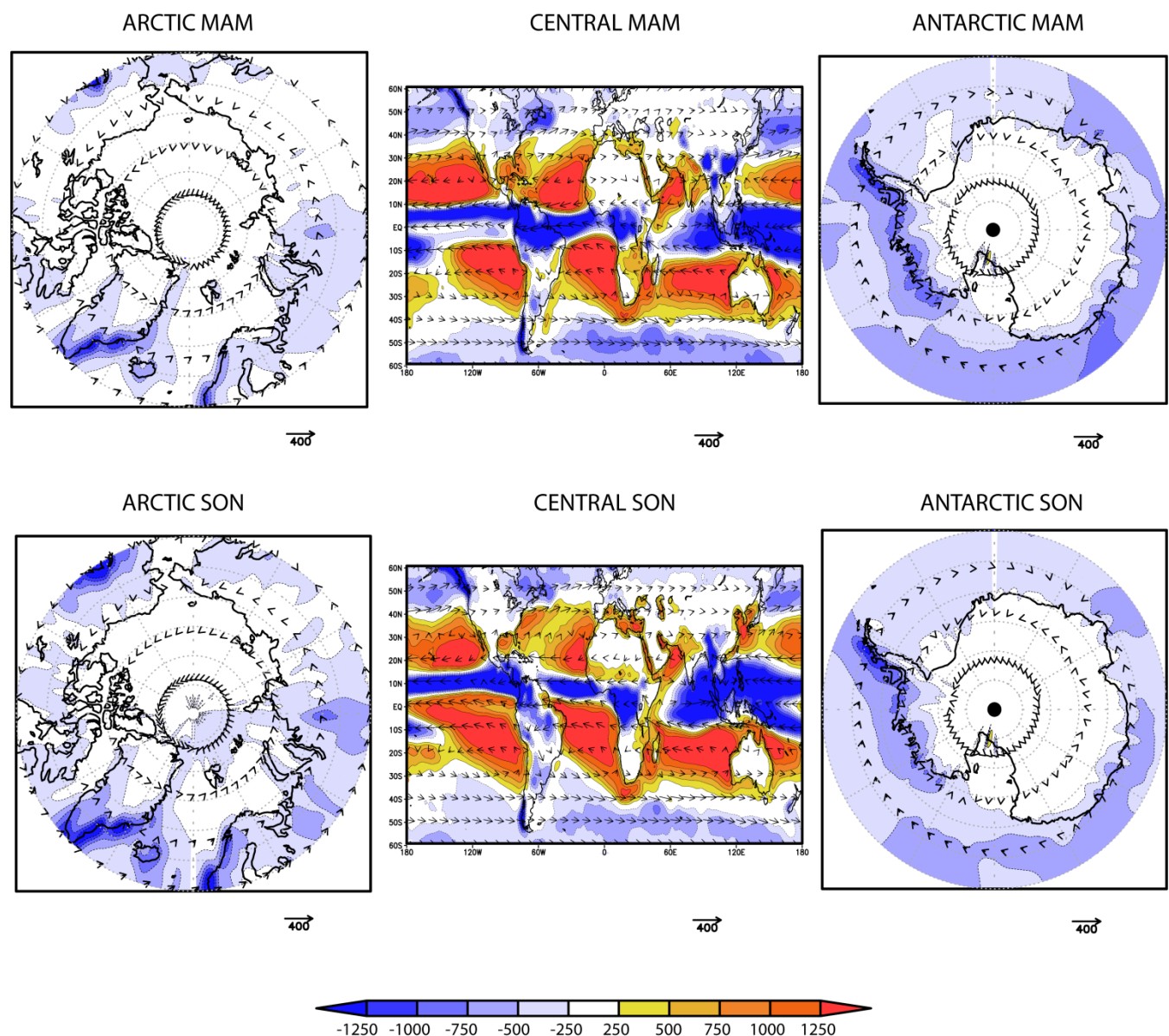

**Figure S1**: 1980-2012 vertically integrated moisture flux (vector, kg/m/s) and its respective divergence (shaded, mm/yr) in the Arctic (left-hand column), the Central (central column), and the Antarctic (right-hand column) domains for the MAM (top row), and SON (bottom row) temporal means. Data: Era-Interim

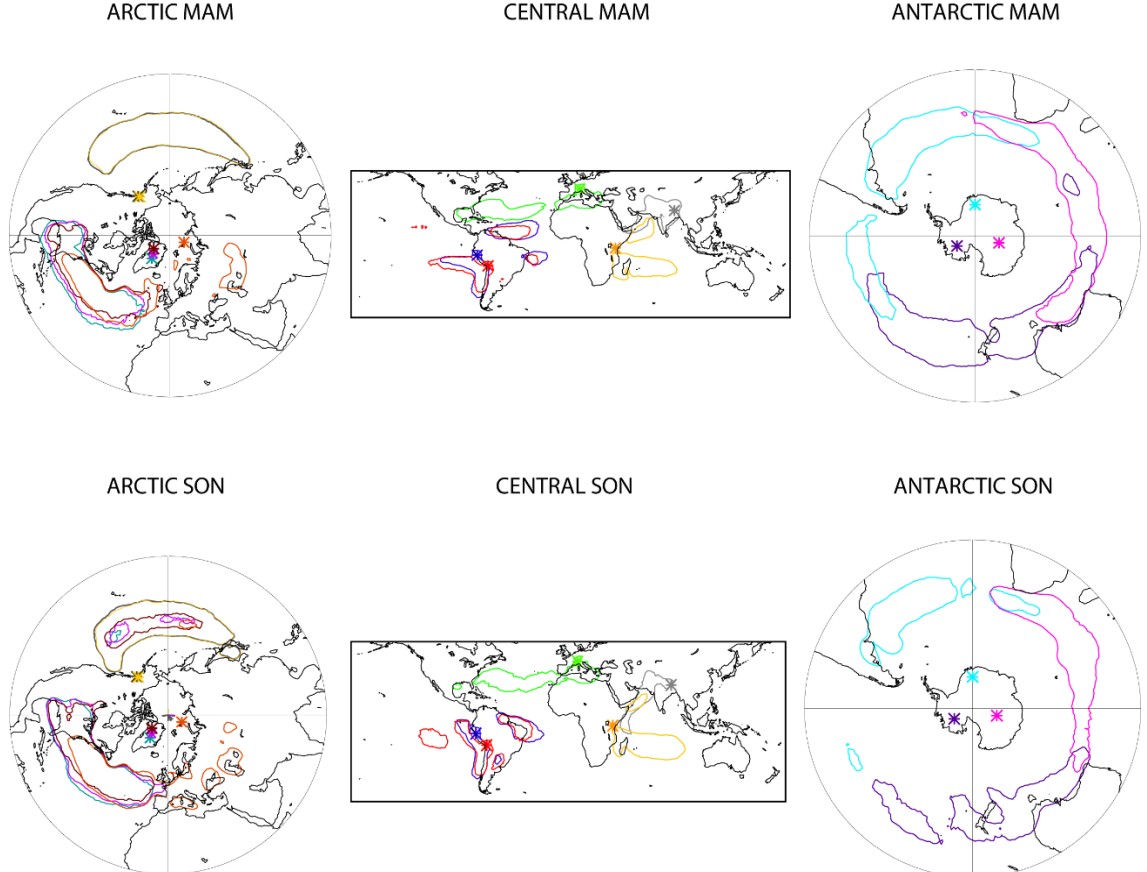

**Figure S2:** Moisture sources for the ice–core sites located in the Arctic (left-hand column), Central (central column), and Antarctic (right-hand column) domains for the MAM (top line), and SON (bottom line) temporal means. The contour lines represent the 95th percentile of the positive $E$-$P$ values in the annual mean for each ice-core Lagrangian analysis. The asterisks indicate the location of the ice-core sites investigated, represented by different colours. The colours are for the Arctic domain: Logan (gray), Bona-Churchill (yellow), Windy Dome (orange), GISP-2 (blue), NEEM (brown), NGRIP (violet); for the Central domain: Huascarán (dark blue), Sajama (red), Grenzgletscher (green), Kilimanjaro (yellow), Everest (gray); for the Antarctic Domain: Byrd (purple), DML (clear blue), and Vostok (magenta).