# Peer review of "A Lagrangian analysis of the present-day sources of moisture for major ice-core sites"

_Earth System Dynamics, 2015_

## Referee Comment (RC1) · Anonymous Referee #1 · 16 Feb 2016

This is an informative contribution to the body of the literature on the moisture sources for ice-core sites. The authors implement a lagrangian approach (the FLEXPART model and ERA-Interim data) to identify the moisture sources of fourteen ice-core sites. From the point of view of this reviewer the results are interesting and useful for being exploited in the future, especially through the comparison with isotopic signals. On the other hand the paper is well structured and is presented in a clear form. I recommend to accept the paper for publication with minor revisions.

Specific comments

*Some ice core sites are widely known (as Vostok) but not all of them, at least for this reviewer. Could the authors add a bibliographic reference for the data of table 1?.

*The applicability of the method for computing moisture sources has been widely

demonstrated in many regions around the world, even for reduced regions. In this work, besides the fact that the horizontal resolution is 1° and that there are areas with low data density (as Antarctica), the sites are generally located in high altitudes. How many particles are typically found in these fourteen locations?.

Technical corrections

*In page 8 (summary), lines 1-2 and lines 7-9 are repeated.

---

## Referee Comment (RC2) · Anonymous Referee #2 · 29 Feb 2016

The paper presents an analysis of moisture sources for a number of ice core sites based on a backward trajectory analysis in ERA-Interim data. The paper provides some interesting results, which eventually may become relevant for interpreting proxies. The results are nicely described and the paper seems scientifically sound. However, for a person not familiar with the methodology (like myself) the description of the method is not easy to follow and needs to be improved. Whether this is major or minor I leave up to the editor.

Main point The method was not clear to me at all. The abstract speaks about backward trajectories, but the more I read I assumed these were forward trajectories analysed backward, but I am not sure. Furthermore, it is not clear to me whether the frequency distribution of particles is accounted for or not. The 2 million particles may be evenly distributed initially, but the sampling procedure must introduce large differences in the
density of trajectories. For instance, for the sampled particles, "summing up" (e-p) in the vertical will not give (E-P) at the surface (because there may be layers in between whose air travels in a different direction and will not reach the ice core site). Furthermore, it seems that the vertical dimension is neglected. Let's assume a (climatological) circulation such that, when your trajectories pass over a location where evaporation usually takes place, the trajectories pass mostly at high altitudes and the particles do not actually pick up a lot of moisture whereas the moist lower layers rain out upon reaching the first mountain chain and never reach the ice core site. Wouldn't this matter? The paragraph on P. 2 to 3 on dq/dt implies that you are tracking moisture changes along a trajectory, but the further I read I think you do not. Rather, it seems that you overlay the position of the particles onto a field of E-P which you could have obtained as well from a Eulerian approach (precipitable water tendency plus vertically integrated moisture flux divergence). I admit I have not read Gimeno et al. (2012), but I think the reader should be given more information here. Below are some more detailed comments P. 3. L. 2: "By summing (e-p)..." Do you mean integrating? (I anyway struggle with the units here). Is there any control (e.g., weighting) of the vertical distribution or is it sampled well enough that this is not necessary? Do you need the vertical distribution at all? P. 3. L. 9: Here's probably where my confusion starts. Since these are backward trajectories (as mentioned in the abstract) you do not need 2 million but only those that arrive at the ice core locations, right? Or are trajectories calculated forward but then the analysis treats them as backward trajectories? Furthermore, since you track the particles for 10 days, how often is the model re-initialised? If it is only initialised once and then everything else is done by sampling, I think it needs to be stated that the trajectories provide a good sample. For instance, after 30 years of simulation all particles might have ended up in the subtropical jet and stay there. Conversely, is the number of particles above a given ice core site the same for all time steps? If not, do you weigh the results somehow? Perhaps I am just ignorant and perhaps this approach is so well known that no further explanations are required, but as a reader I would be glad for some help. Otherwise I think this is a fascinating paper. P. 3. L.

11: Are retro-trajectories backward trajectories? P. 4, L. 6: What are "target areas": Are these areas where particles end after a 10-day period or an area over which they pass at any time during a 10-day period? P. 4, L. 10: "backward methodology": forward trajectories analysed backward? P. 4, L. 11: "(E-P > 0)" Now I am confused. Why not "(e-p > 0)"? If the assumption is that each particle (i.e., e-p) behaves in the same way as the integrated column (i.e., E-P), then there would be no need to track moisture at all. Simply use (Eulerian) E-P from ERA-Interim and the position of the particle from FLEXPART. Is that what you do? So why do you initially integrate (e-p) rather than taking it directly from ERA-Interim? P. 4, L. 13: "E-P averaged over the whole tracking period (10 days)" Again, I understand this such that you basically use a (Eulerian) map of E-P and then sample it at the locations and time where air parcels pass it (at any altitude) and then reach the ice-core site within at most 10 days. But you only do that if E-P is positive. It seems that there is no weighting that accounts for unequal distribution for particles. But this should be stated (and justified). I am still confused, though, why you need e-p at all. P. 4, L. 13: "95th percentile" Do I understand this correctly: For each ice core you would show the 5% grid points with the highest E-P (if you showed the annual mean)? P. 4, L. 32: The VIMF is calculated directly from ERA-Interim, right? Or is it from the Lagrangian approach? P. 5, L. 1: The difference between vertically integrated moisture flux divergence and E-P is the tendency in precipitable water (i.e., storage), which can be neglected over long time periods. So the E-P figure would look identical, right?

Minor points The first paragraph of the introduction is rather vague as to the processes causing changes in isotope abundances in ices cores: they depend on "local conditions," changes in "relative moisture of the source" (what is that?), "changes in these source conditions," etc. It would be good to be more specific P. 2. L. 29: Give a reference for MERRA. P. 2, L. 19: "unrealistic fluctuations in humidity can be taken as moisture fluxes": I do not understand that. P. 3. L. 23: "low data density": Do you mean the number of particles or the quality of the reanalysis there. P. 10, l. 16: Anchmann -> Auchmann P. 11, last reference: Chack names. P. 13, Table 1: couverture -> coverage

Fig. 2: The arrows do not help very much; perhaps make them larger

[Figure]

---

## Referee Comment (RC3) · Anonymous Referee #3 · 7 Mar 2016

This paper proposes to identify the moisture sources for 14 ice cores divided into 3 large "domains" (Arctic, Central and Antarctic). The findings are as expected, that the subtropical oceans provide most moisture (although the contributions change through the year).

In the Introduction on the discussion on D-excess there is no mention of air moisture trajectory history as a control, ie the D-excess will change when moisture moves over dry as opposed to wet land for example (a major influence in the Central domain?). Should trajectory history be added?

In the second paragraph (and the title) the authors mention the Lagrangian diagnostic scheme but there is no explanation of this for the non-expert.

The authors follow a previously published approach from about a decade ago, which

was updated in a 2014 paper, but they do not consider the updates. It wasn't really clear to me why they didn't apply the updated method.

Overall they identify moisture source areas which are as expected or have previously been identified from trajectory studies. I wonder if they can ground truth some of their findings from data in the literature as this purely modelling approach seems deficient when so much observational data is available?

---

## Editor Comment (EC1) · M. Crucifix (Editor) · 7 Mar 2016

Dear authors,

Thank you for having submitted your work to Earth System Dynamics. We have now received three reviews of your manuscript. While they are generally concise and positive about the interest of the work, two reviewers advise substantial revisions of your article, mainly calling for clarification of the methods. This is important because ESD aims at multi-disciplinarity. It is therefore important that the merits of the methodology over other approaches be clear to a large enough audience.

The article is also in need of more editorial work. Style is casual at places, and there is scope for improvement in style, choice of words and grammar.

---

## Author Comment (AC1) · 15 Apr 2016

We would like to thank the commentaries of the Editor and the three reviewers. We totally agree that the description of method must be clarified in order to illustrate the potential use of the technique. For this purpose, the section "data and methods" was re-written taking into consideration all questions arisen in order to provide a detailed description of the approach. We hope to answer all questions arisen by the reviewers. In particular, the commentaries arisen by the Reviewer 2 have been re-arranged focusing on the different themes.

---

## Author Comment (AC2) · 15 Apr 2016

Please, read our answer for your commentaries. Thank you very much for your review.

*Some ice core sites are widely known (as Vostok) but not all of them, at least for this reviewer. Could the authors add a bibliographic reference for the data of table 1? - We included some examples of studies concerning the ice core sites investigated in the present work. The references were added in the Table 1 of the manuscript.

GISP-2: Meese, D. A., Gow, A. J., Alley, R. B., Zielinski, G. A., Grootes, P. M., Ram, M., Taylor, K. C., Mayewski, P. A., and Bolzan, J. F.:The Greenland Ice Sheet Project 2 depth-age scale: Methods and results, J. Geophys. Res., 102(C12), 26411–26423, doi:10.1029/97JC00269, 1997.

[Figure]

NGRIP: Andersen, K. K., Azuma, N., Barnola, J.-M., Bigler, M., Biscaye, P., et al.: High-resolution record of Northern Hemisphere climate extending into the last interglacial period, Nature, 431, 147-151, doi:10.1038/nature02805, 2004

NEEM: Rasmussen, S.O., Abbott, P.M., Blunier, T., Bourne, A.J., Brook, E., et al.: A first chronology for the North Greenland Eemiam Ice Drilling (NEEM) ice core. Clim. Past, 2713-2730, doi: 10.5194/cp-9-2713-2013, 2013.

Monte Logan: Fisher, D.A., Wake, C., Kreutz, K., Yalcin, K., Steig, E., et al.: Stable isotope records from Mount Logan, Eclipse Ice Cores and Nearby Jellybean Lake. Water cycle of the North Pacific over 2000 years and over five vertical kilometres: sudden shift and tropical connections, Geographie Physique et Quaternaire, 58, 2-3, 337-352, 2004.

Bona-Churchill: Mashiotta, T.A., Thompson, L. G. and Davis, M. E.: The White River Ash: New evidence from the Bona-Churchill ice core record, Eos Trans. AGU, 85(47), Fall Meet. Suppl., Abstract PP21A-1369, 2004.

Wind Dome: Henderson, K. A.: An ice core paleoclimate study of Windy Dome, Franz Josef Land (Russia): development of a recent climate history for the Barents Sea. Diss. The Ohio State University, 2002.

Huascaran: Thompson, L.G., Mosley-Thompson, E., Davis, M.E., Lin, P-N., Henderson, K.A., Cole-Dai, J., Bolzan, J.F. and Liu, K-b.: Late Glacial Stage and Holocene tropical ice core records from Huascaran, Peru, Science, 269, 46-50, 1995.

Sajama: Thompson, L.G., Davis, M.E., Thompson, E.M., Sowers, T.A., Henderson, K.A., Zagorodnov, V.S., Lin, P.N., Mikhalenko, V.N., Campen, R.K., Bolzan, J.F., Cole-Dai, J. and Francou, B.: A 25,000 year tropical climate history from Bolivian ice cores, Science, 282(5295), 1858-1864, 1998.

Grenzgletscher: Eichler, A., Schwikowski, M., Gäggeler, H. W., Furrer, V, Synal, H.-A., Beer, J., Saurer, M., and Funk, M.: Glaciochemical dating of an ice

core from upper Grenzgletscher (4200 m a.s.l.), J. Glaciol., 46, 507–515, doi: 10.3189/172756500781833098, 2000.

Everest: Hou, S., Chappellaz, D. Raynaud, Masson-Delmotte, V., Jouzel, J., Bousquet, P. and Hauglustaine, D.: A new Himalayan ice core $CH_4$ record: possible hints at the preindustrial latitudinal gradient, Clim. Past, 9, 2549-2554, Doi: 10.5194/cp-9-2549-2013, 2013.

Kilimanjaro: Thompson, L. G., Mosley-Thompson, E., Davis, M. E., Henderson K. A., Brecher, H. H., Zagorodnov, V. S., Mashiotta, T. A., Lin, P.-N., Mikhalenko, V. N., Hardy, D. R. and Beer, J.: Kilimanjaro Ice core Records: Evidence of Holocene climate change in Tropical Africa, Science, 298, 5593, 589-593, DOI: 10.1126/science 1073198, 2002.

Byrd: Thompson, L.G., Hamilton, W.L. and Bull, C.: Climatological implications of microparticle concentrations in the ice core from Byrd Station, Western Antarctica, Journal of Glaciology, 14(72), 433-444, 1975.

Epica DML: Ruth, U., Barnola, J.M., Beer, J., Bigler, M., Blunier, T., Castellano, E., Fischer, H., Fundel, F., Huybrechts, P., Kaufmann, P., Kipfstuhl, S., Lambrecht, A., Morganti, A., Oerter, H., Parrenin, F., Rybak, O., Severi, M., Udisti, R., Wilhelms, F. and Wolff, E.: "EDML1": a chronology for the EPICA deep ice core from Dronning Maud Land, Antarctica, over the last 150000 years, Clim. Past, 3, 475-484, 2007.

Vostok: Petit, J.R., Jouzel, J., Raynaud, D., Barkov, N.I., Barnola, J.-M., et al.: Climate and atmospheric history of the past 420,000 years from the Vostok ice core, Antarctica, Nature, 399, 429-436, 1999.
* * *
*The applicability of the method for computing moisture sources has been widely demonstrated in many regions around the world, even for reduced regions. In this work, besides the fact that the horizontal resolution is 1o and that there are areas with low data density (as Antarctica), the sites are generally located in high altitudes. How

many particles are typically found in these fourteen locations? - The approximate number of particles found per time step over each of the fourteen locations was included in the table 1 of the manuscript. Stohl and James (2004) state that the estimative of the moisture budget is valid when the number of particles per grid column of the input meteorological data exceeds the number of the layers.

\*\*\*\*\*\*\*\*\*\*\*\*\*\*\*\*\*\*\*\*\*\*\*\*\*\*\*\*\*\*\*\*\*\*\*\*\*\*\*\*\*\*\*\*\*\*\*\*\*\*\*\*\*\*\*\*\*\*\*\*\*\*\*\*\*\*\*\*\*\*\*\*\*\*\*\*\*\*\*\*\*\*\*

Technical corrections \*In page 8 (summary), lines 1-2 and lines 7-9 are repeated. -Thank you. Lines 1-2 were removed from the text.

Please also note the supplement to this comment:
http://www.earth-syst-dynam-discuss.net/esd-2015-97/esd-2015-97-AC2-supplement.pdf

**Supplement:**

| Domain | Num. | Site | Lat. | Long. | High (m) | Max. depth (m) | Temporal coverage (year) | Number ~ of particles identified per time step | Reference |
|---|---|---|---|---|---|---|---|---|---|
| Arctic | 1 | GISP2 | 72,60°N | 38,50°W | 3200 | ~2790 | ~110000 | 220 | Meese et al. (1997) |
| | 2 | NEEM | 77,45°N | 51,07°W | 2479 | ~2540 | ~108000 | 185 | Rasmussen et al. (2013) |
| | 3 | NGRIP | 75,10°N | 42,30°W | 2917 | ~3084 | ~123000 | 185 | Andersen et al. (2004) |
| | 4 | MONTE LOGAN | 60,58°N | 140,58°W | 5340 | ~186 | ~8000 | 230 | Fisher et al. (2004) |
| | 5 | BONA-CHURCHILL | 61,40°N | 141,70°W | 4420 | ~460 | ~2500 | 215 | Mashiotta et al. (2004) |
| | 6 | WINDY DOME | 80,78°N | 63,53°E | 580 | ~315 | ~10000 | 780 | Henderson (2002) |
| Central | 7 | HUASCARÁN | 9,18°S | 78,02°W | 6048 | ~166 | ~20000 | 75 | Thompson et al. (1995) |
| | 8 | SAJAMA | 18,10°S | 68,97°W | 6542 | ~133 | ~20000 | 66 | Thompson et al. (1998) |
| | 9 | GRENZGLETSCHER | 45,92°N | 7,87°E | 4200 | ~125 | ~77-20 | 150 | Eichler et al. (2000) |
| | 10 | EVEREST | 28,02°N | 86,97°E | 6518 | ~117 | ~4000 | 70 | Hou et al. (2013) |
| | 11 | KILIMANJARO | 3,13°S | 37,58°E | 5893 | ~51 | ~11700 | 80 | Thompson et al. (2002) |
| Antarctic | 12 | BYRD | 80°S | 119°W | 1530 | ~2164 | ~100000 | 450 | Thompson et al. (1975) |
| | 13 | EPICA DML | 75°S | 0°E | 2892 | ~2774 | ~150000 | 1215 | Ruth et al. (2007) |
| | 14 | VOSTOK | 78°S | 106°E | 3488 | ~3623 | ~440000 | 790 | Petit et al. (1999) |

---

## Author Comment (AC3) · 15 Apr 2016

We would like to thank you for your review. Your commentaries have been re-arranged focusing on the different questions arisen. Totally agreeing that the methodology was not explained in the original version in details, this section has been re-writen in this revised version focusing on answering your questions.

\*\*\*\*\*\*\*\*\*\*\*\*\*\*\*\*\*\*\*\*\*\*\*\*\*\*\*\*\*\*\*\*\*\*\*\*\*\*\*\*\*\*\*\*\*\*\*\*\*\*\*\*\*\*\*\*\*

*The method was not clear to me at all. * The abstract speaks about backward trajectories, but the more I read I assumed these were forward trajectories analysed backward, but I am not sure. *Or are trajectories calculated forward but then the analysis treats them as backward trajectories? * P. 4, L. 10: "backward methodology": forward trajectories analysed backward?

[Figure]

-The method consists in analysing the 10-d backward trajectories of the particles identified over the 14 ice core sites studied. These particles are advected backward in time using three-dimensional wind taken from the meteorological data (e.g. reanalysis project) every time step.

\*\*\*\*\*\*\*\*\*\*\*\*\*\*\*\*\*\*\*\*\*\*\*\*\*\*\*\*\*\*\*\*\*\*\*\*\*\*\*\*\*\*\*\*\*\*\*\*\*\*\*\*\*\*\*\*\*\*\*\*\*\*\*\*\*\*\*\*\*\*\*\*\*\*\*\*\*\*\*\*

\* Furthermore, it is not clear to me whether the frequency distribution of particles is accounted for or not. The 2 million particles may be evenly distributed initially, but the sampling procedure must introduce large differences in the density of trajectories. \* P. 3. L. 2: "By summing (e-p)..." Do you mean integrating? (I anyway struggle with the units here). Is there any control (e.g., weighting) of the vertical distribution or is it sampled well enough that this is not necessary? Do you need the vertical distribution at all? \* Since these are backward trajectories (as mentioned in the abstract) you do not need 2 million but only those that arrive at the ice core locations, right? \* Conversely, is the number of particles above a given ice core site the same for all time steps? If not, do you weigh the results somehow?

- The FLEXPART data set used in this study was provided by a global experiment in which the entire global atmosphere was divided into approximately 2.0 million 'particles', and the number of particles per time step was kept constant along the analysis. The approximate number (there is a small variation) of particles found per time step over each of the fourteen ice core sites was included in the table 1 of the manuscript. Stohl and James (2004) state that the estimative of the moisture budget is valid when the number of particles per grid column of the input meteorological data exceeds the number of the layers. The analysis was based on the particles that arrive at the ice core locations.

\*\*\*\*\*\*\*\*\*\*\*\*\*\*\*\*\*\*\*\*\*\*\*\*\*\*\*\*\*\*\*\*\*\*\*\*\*\*\*\*\*\*\*\*\*\*\*\*\*\*\*\*\*\*\*\*\*\*\*\*\*\*\*\*\*\*\*\*\*\*\*\*\*\*\*\*\*\*\*\*

\* For instance, for the sampled particles, "summing up" (e-p) in the vertical will not give (E-P) at the surface (because there may be layers in between whose air travels in a

different direction and will not reach the ice core site). * The paragraph on P. 2 to 3 on dq/dt implies that you are tracking moisture changes along a trajectory, but the further I read I think you do not. Rather, it seems that you overlay the position of the particles onto a field of E-P which you could have obtained as well from a Eulerian approach (precipitable water tendency plus vertically integrated moisture flux divergence). I admit I have not read Gimeno et al. (2012), but I think the reader should be given more information here. * P. 4, L. 11: "(E-P > 0)" Now I am confused. Why not"(e-p > 0)"? If the assumption is that each particle (i.e., e-p) behaves in the same way as the integrated column (i.e., E-P), then there would be no need to track moisture at all. Simply use (Eulerian) E-P from ERA-Interim and the position of the particle from FLEXPART. Is that what you do? So why do you initially integrate (e-p) rather thanctaking it directly from ERA-Interim? * P. 5, L. 1: The difference between vertically integrated moisture flux divergence and E-P is the tendency in precipitable water (i.e., storage), which can be neglected over long time periods. So the E-P figure would look identical, right? *P. 4, L. 13: "E-P averaged over the whole tracking period (10 days)" Again, I understand this such that you basically use a (Eulerian) map of E-P and then sample it at the locations and time where air parcels pass it (at any altitude) and then reach the ice-core site within at most 10 days. But you only do that if E-P is positive. It seems that there is no weighting that accounts for unequal distribution for particles. But this should be stated (and justified). I am still confused, though, why you need e-p at all.

- We have detailed in more depth the methodology in the manuscript. By summing (e-p) for all the particles residing in the atmospheric column over a given area A, we obtained the surface freshwater flux (E-P). If we consider all the particles present in the atmospheric column, the results would be similar to the freshwater flux calculated via the Eulerian reference (Stohl and James, 2004). Nevertheless, the Lagrangian methodology allows us to identify the particles affecting a particular region and to calculate the surface freshwater flux (E-P) using information on the trajectories of these selected particles. Figures 3 and S2 of the manuscript are based on (E-P) of the tracked particles averaged over the 10-d period and redistributed on a regular 1° grid.

These points were clarified in the new version of the manuscript.

\*\*\*\*\*\*\*\*\*\*\*\*\*\*\*\*\*\*\*\*\*\*\*\*\*\*\*\*\*\*\*\*\*\*\*\*\*\*\*\*\*\*\*\*\*\*\*\*\*\*\*\*\*\*\*\*\*\*\*\*\*\*\*\*\*\*\*\*\*\*\*\*\*\*\*\*

\* Furthermore, it seems that the vertical dimension is neglected. Let's assume a (climatological) circulation such that, when your trajectories pass over a location where evaporation usually takes place, the trajectories pass mostly at high altitudes and the particles do not actually pick up a lot of moisture whereas the moist lower layers rain out upon reaching the first mountain chain and never reach the ice core site. Wouldn't this matter?

- The vertical dimension is taken into account in this approach. In the model the atmosphere is divided homogeneously into three-dimensional finite elements (hereafter 'particles'), each representing a fraction of the total atmospheric mass (Stohl and James, 2004). These particles may be advected backward or forward in time using three-dimensional wind taken from the Era-Interim. This issue has also been detailed in the revised manuscript.

\*\*\*\*\*\*\*\*\*\*\*\*\*\*\*\*\*\*\*\*\*\*\*\*\*\*\*\*\*\*\*\*\*\*\*\*\*\*\*\*\*\*\*\*\*\*\*\*\*\*\*\*\*\*\*\*\*\*\*\*\*\*\*\*\*\*\*\*\*\*\*\*\*\*\*\*

\*Furthermore, since you track the particles for 10 days, how often is the model re-initialised? If it is only initialized once and then everything else is done by sampling, I think it needs to be stated that the trajectories provide a good sample. For instance, after 30 years of simulation all particles might have ended up in the subtropical jet and stay there.

- The particles are advected backward (or forward) in time using data taken from Era-Interim every time step.

\*\*\*\*\*\*\*\*\*\*\*\*\*\*\*\*\*\*\*\*\*\*\*\*\*\*\*\*\*\*\*\*\*\*\*\*\*\*\*\*\*\*\*\*\*\*\*\*\*\*\*\*\*\*\*\*\*\*\*\*\*\*\*\*\*\*\*\*\*\*\*\*\*\*\*\*

\* P. 3. L. 11: Are retro-trajectories backward trajectories?

- Yes. However, the term was replaced by backward trajectories in order to avoid any

misunderstanding.

\*\*\*\*\*\*\*\*\*\*\*\*\*\*\*\*\*\*\*\*\*\*\*\*\*\*\*\*\*\*\*\*\*\*\*\*\*\*\*\*\*\*\*\*\*\*\*\*\*\*\*\*\*\*\*\*\*\*\*\*\*\*\*\*\*\*\*\*\*\*\*\*\*\*\*\*\*\*\*\*\*

\* P. 4, L. 6: What are "target areas": Are these areas where particles end after a 10-day period or an area over which they pass at any time during a 10-day period?

- In this study the ice core sites are the target areas. Our study is based on backward in time trajectories of the particles identified over the ice core sites, the area where the particles end is the ice core site in day 0, and the previous days are day-1, day-2,. . . day-10. This explanation was added in the new version of the manuscript.

\*\*\*\*\*\*\*\*\*\*\*\*\*\*\*\*\*\*\*\*\*\*\*\*\*\*\*\*\*\*\*\*\*\*\*\*\*\*\*\*\*\*\*\*\*\*\*\*\*\*\*\*\*\*\*\*\*\*\*\*\*\*\*\*\*\*\*\*\*\*\*\*\*\*\*\*\*\*\*\*\*

\* P. 4, L. 13: "95th percentile" Do I understand this correctly: For each ice core you would show the 5% grid points with the highest E-P (if you showed the annual mean)?

- Yes, the 95th percentile criteria would show the 5% grid points with the highest positive (E-P) values in the annual mean map obtained for each ice core site. This explanation was added in the manuscript..

\*\*\*\*\*\*\*\*\*\*\*\*\*\*\*\*\*\*\*\*\*\*\*\*\*\*\*\*\*\*\*\*\*\*\*\*\*\*\*\*\*\*\*\*\*\*\*\*\*\*\*\*\*\*\*\*\*\*\*\*\*\*\*\*\*\*\*\*\*\*\*\*\*\*\*\*\*\*\*\*\*

\* P. 4, L. 32: The VIMF is calculated directly from ERA-Interim, right? Or is it from the Lagrangian approach?

- The VIMF is calculated directly from ERA-Interim. It was clarified in the text and in the respective figure captions.

\*\*\*\*\*\*\*\*\*\*\*\*\*\*\*\*\*\*\*\*\*\*\*\*\*\*\*\*\*\*\*\*\*\*\*\*\*\*\*\*\*\*\*\*\*\*\*\*\*\*\*\*\*\*\*\*\*\*\*\*\*\*\*\*\*\*\*\*\*\*\*\*\*\*\*\*\*\*\*\*\*

\* The first paragraph of the introduction is rather vague as to the processes caus-ing changes in isotope abundances in ices cores: they depend on "local conditions," changes in "relative moisture of the source" (what is that?), "changes in these source conditions," etc. It would be good to be more specific

- The first paragraph was re-written in order to clarify the points arisen by the Reviewer and the new version is below. The reference Merlivat and Jouzel (1979) has been replaced by a more recent one (Jouzel et al., 2013).

"The most successful reconstruction of past climate has been due to the fact that stable water isotopes are conserved in ice cores (e.g. Jouzel et al., 1982; Dansgaard et al., 1993). The isotopic composition of precipitation, in deuterium, oxygen-18 and oxygen-17, depends on the climatic conditions prevailing in the oceanic regions where it originates (i.e. the sources), mainly the sea surface temperature and the relative humidity of air (Jouzel et al., 2013). The deuterium excess, for example, may be seen as a control parameter of air moisture trajectory history, because it will change when the trajectory moves over regions presenting different moisture conditions (e.g. sea/land, dry/wet land).Deuterium excess variations have been traditionally associated to changes in the temperature of the oceanic sources, but nowadays it is thought to be also related with changes in the relative humidity of the air in the source region (Pfahl and Sodemann, 2014). In any case, deuterium excess variations in ice cores may reflect past changes in the climate conditions of the oceanic sources (e.g. Masson-Delmotte et al., 2005; Steffensen et al., 2008). This information can be very useful to understand changes linked to modifications in the atmospheric circulation because the position and conditions of the moisture sources for precipitation could be altered (e.g. Masson-Delmotte et al., 2005). That is why the knowledge on the transport of moisture is crucial for the interpretation of stable isotopes in precipitation and in paleo-archives through the understanding of the physical climatic processes involved (Sodemann and Zubler, 2009)."

Jouzel, J., Delaygue, G., Landais, A., Masson-Delmotte, V., Risi, C., and Vimeux, F.: Water isotopes as tools to document oceanic sources of precipitation. Water Resour. Res., 49, 7469–7486, doi:10.1002/2013WR013508, 2013.
* * *
* P. 2. L. 29: Give a reference for MERRA.

- The reference was included: Rienecker, M.M., Suarez, M.J, Gelaro, R., Todling, R., Bacmeister, J., et al: MERRA: NASA's Modern-Era Retrospective Analysis for Research and Applications. J. Climate, 24, 3624–3648. doi: http://dx.doi.org/10.1175/JCLI-D-11-00015.1, 2011
* * *
* P. 2, L. 19: "unrealistic fluctuations in humidity can be taken as moisture fluxes": I do not understand that.

- When applying a time derivative of the humidity, the numerical errors associated with the temporal variations in the moisture present in a particle can be taken as moisture fluxes. In consequence, if the reanalysis data used to drive the method do not properly close the water budget, then the method may suffer from considerable inaccuracies(Gimeno et al., 2012).
* * *
* P. 3. L. 23: "low data density": Do you mean the number of particles or the quality of the reanalysis there.

- We refer to regions with low observational data coverage. It was explained in the text.
* * *
* P. 10, l. 16: Anchmann -> Auchmann Done

* P. 11, last reference: Chack names. Corrected. Thanks.

* P. 13, Table 1: couverture -> coverage Done. Thanks.

* Fig. 2: The arrows do not help very much; perhaps make them larger Done.
* * *

---

## Author Comment (AC4) · 15 Apr 2016

Thank you very much for your review. Please, read our answer to your commentaries below.

\* In the Introduction on the discussion on D-excess there is no mention of air moisture trajectory history as a control, ie the D-excess will change when moisture moves over dry as opposed to wet land for example (a major influence in the Central domain?). Should trajectory history be added?

- This statement has been mentioned in the first paragraph of the Introduction.

\*\*\*\*\*\*\*\*\*\*\*\*\*\*\*\*\*\*\*\*\*\*\*\*\*\*\*\*\*\*\*\*\*\*\*\*\*\*\*\*\*\*\*\*\*\*\*\*\*\*\*\*\*\*\*\*\*\*\*\*\*\*

\* In the second paragraph (and the title) the authors mention the Lagrangian diagnostic

scheme but there is no explanation of this for the non-expert.

- We totally agree that the explanation of the method would be more detailed for in order to illustrate the potential use of the technique for the scientific community. Please, read the answers for the Reviewer 2 and the re-written version of the "data and methods" section.

\*\*\*\*\*\*\*\*\*\*\*\*\*\*\*\*\*\*\*\*\*\*\*\*\*\*\*\*\*\*\*\*\*\*\*\*\*\*\*\*\*\*\*\*\*\*\*\*\*\*\*\*\*\*\*\*\*\*\*\*\*\*\*\*\*\*\*\*\*\*\*

\* The authors follow a previously published approach from about a decade ago, which was updated in a 2014 paper, but they do not consider the updates. It wasn't really clear to me why they didn't apply the updated method.

- The methodology applied here follows the pioneers works of Stohl and James (2004; 2005) simply considering the regions of (E-P) >0 as moisture sources and tracking all the air masses reaching the target region, being or not associated with precipitation events. Other moisture sources diagnostic schemes are available (Gimeno et al.2012), such as the Lagrangian method proposed by Sodemann et al. (2008) to identify the origin of precipitation. In their approach, the cumulative moisture changes along the trajectory are also considered besides the net gain or loss at each grid point, what is necessary for quantifying the contribution of the air parcel for the precipitation in the target region. Anyway, since the purpose of the present work is to estimate the climatological moisture sources of all air masses reaching the target regions, independently of the occurrence of precipitation in the ice core sites, we believe that the use of this simple Lagrangian approach is reasonable. This discussion was included in the last paragraph of the "Data an Methods" section.

\*\*\*\*\*\*\*\*\*\*\*\*\*\*\*\*\*\*\*\*\*\*\*\*\*\*\*\*\*\*\*\*\*\*\*\*\*\*\*\*\*\*\*\*\*\*\*\*\*\*\*\*\*\*\*\*\*\*\*\*\*\*\*\*\*\*\*\*\*\*\*

\* Overall they identify moisture source areas which are as expected or have previously been identified from trajectory studies. I wonder if they can ground truth some of their findings from data in the literature as this purely modelling approach seems deficient

when so much observational data is available?

- Two references published in the last five years have been added in order to provide comparison with up-to-dated works. Anyway, the findings of both methods are not considering exactly the same climatic conditions and the comparison between them must be done cautiously. On one hand, as explained in the previous question, our method tracks all the air masses reaching the target region (being or not associated with precipitation events). On the other hand, the results based on observational data imply in investigating the origin of vapor associated with precipitation episodes in the ice-core sites (and these specific synoptic situations).

Yao, T., et al. (2013), A review of climatic controls on $\delta$18O in precipitation over the Tibetan Plateau: Observations and simulations, Rev. Geophys., 51, 525–548, doi:10.1002/rog.20023.

Kurita, N. (2011), Origin of Arctic water vapor during the ice-growth season, Geophys. Res. Lett., 38, L02709, doi:10.1029/2010GL046064.